# Positive Effects of Argon Inhalation After Traumatic Brain Injury in Rats

**DOI:** 10.3390/ijms252312673

**Published:** 2024-11-26

**Authors:** Viktoriya V. Antonova, Denis N. Silachev, Egor Y. Plotnikov, Irina B. Pevzner, Mikhail E. Ivanov, Ekaterina A. Boeva, Sergey N. Kalabushev, Mikhail Ya. Yadgarov, Rostislav A. Cherpakov, Oleg A. Grebenchikov, Artem N. Kuzovlev

**Affiliations:** 1Federal Research and Clinical Center of Intensive Care Medicine and Rehabilitology, Moscow 107031, Russia; eboeva@fnkcrr.ru (E.A.B.); skalabushev@fnkcrr.ru (S.N.K.); mikhail.yadgarov@mail.ru (M.Y.Y.); zealot333@mail.ru (R.A.C.); oleg.grebenchikov@yandex.ru (O.A.G.); artem_kuzovlev@mail.ru (A.N.K.); 2A.N. Belozersky Institute of Physico-Chemical Biology, Lomonosov Moscow State University, Moscow 119992, Russia; silachevdn@genebee.msu.ru (D.N.S.); plotnikov@belozersky.msu.ru (E.Y.P.); irinapevzner@mail.ru (I.B.P.); neokarda@mail.ru (M.E.I.)

**Keywords:** traumatic brain injury, TBI, argon, noble gas, neuroprotection, TNFα, CD68, AKT, Nrf2

## Abstract

The noble gas argon is one of the most promising neuroprotective agents for hypoxic-reperfusion injuries of the brain. However, its effect on traumatic injuries has been insufficiently studied. The aim of this study was to analyze the effect of the triple inhalation of the argon-oxygen mixture Ar 70%/O_2_ 30% on physical and neurological recovery and the degree of brain damage after traumatic brain injury and to investigate the possible molecular mechanisms of the neuroprotective effect. The experiments were performed in male Wistar rats. A controlled brain injury model was used to investigate the effects of argon treatment and the underlying molecular mechanisms. The results of the study showed that animals with craniocerebral injuries that were treated with argon inhalation exhibited better physical recovery rates, better neurological status, and less brain damage. Argon treatment significantly reduced the expression of the proinflammatory markers TNFα and CD68 caused by TBI, increased the expression of phosphorylated protein kinase B (pAKT), and promoted the expression of the transcription factor Nrf2 in intact animals. Treatment with an argon-oxygen breathing mixture after traumatic brain injury has a neuroprotective effect by suppressing the inflammatory response and activating the antioxidant and anti-ischemic system.

## 1. Introduction

Recently, the neuroprotective effects of the noble gas argon have been extensively studied in models of hypoxic-ischemic brain injury [1]. Many studies on in vitro models have shown that the use of argon-containing media has a significant neuroprotective effect under both pre- and postconditioning environments [2,3,4,5]. In in vivo models, most data are available for ischemic brain injury models [5,6,7,8,9,10]. Several studies have already analyzed the efficacy of argon treatment [4,11,12,13] and compared its effects with another “inert” gas, xenon, known as a neuroprotectant and approved for clinical use [14]. However, the efficacy of argon treatment in traumatic brain injury (TBI) has not yet been clearly established, as shown by the contradictory results of a small number of studies [14,15,16,17]. This may be partly because the pathogenesis of brain injury in trauma is markedly different from that in ischemic injury [18] and partly because its reproduction under laboratory conditions encounters a number of limitations related to the humane treatment of laboratory animals and the need for anesthesia, which may weaken the neuroprotective effect of the drugs studied [19]. In one of these studies, 24 h after subarachnoid hemorrhage, exposure to argon was found to increase HIF1α and HO-1 levels, which reached control levels after 72 h and were a factor that reduced neuronal death [20]. In another study, HO-1 levels did not peak until the third day and returned to control levels after 7–15 days [21]. Based on this, our investigation of the potential molecular mechanisms of the neuroprotective effect of argon in TBI [18,22,23] was founded on the known pathogenesis of TBI and potential targets for therapeutic effects, as well as on the already known molecular mechanisms of the neuroprotective effect of argon identified in other models [3,24,25,26,27,28,29].

Regarding the investigation of the molecular mechanisms of the neuroprotective effect of argon in TBI, few studies have investigated the potential molecular mechanisms of the neuroprotective effect of argon in models that are similar to the pathogenesis of TBI [20,21]. Moreover, there is no single in vivo study that combines the clinical efficacy of this treatment after severe traumatic brain injury with the investigation of the molecular mechanisms of its application. The aim of our study was to investigate the effect of three inhalations of an argon-oxygen mixture over a two hour period after open traumatic brain injury on the severity of neurological deficits and the degree of brain damage in rats and to explore the signaling pathways responsible for neuroprotection.

## 2. Results

### 2.1. Mortality and Complications

The mortality rate before randomization (anesthesiological and intraoperative) was 18.9% (n = 7), 5.4% of whom died before surgery (n = 2), during surgery 8.1% (n = 3), and in the first hour after induction of TBI 5.4% (n = 2). The achievement of the endpoint (prolonged lateral motionless position—only breathing movements are noticeable, long-term difficulty breathing—animal’s mouth is open, inability of the animal to eat/drink on its own, prolonged lack of response to external stimuli, reluctance to move, long-term inability to maintain a natural body position, and self-mutilation) was observed only in TBI groups during the late postoperative period (after 7 days). In the TBI + iAr group, it was observed in 9.1% (n = 1 out of 11), in the TBI group, it was observed in 20% (n = 2 out of 10), and in the SO group, there was no achievement of the final humane endpoint.

A total of 24 animals underwent magnetic resonance imaging (MRI) on the 14th day of observation; 6 animals were from the SO group, 8 animals were from the TBI group, and 10 animals were from the TBI + iAr group. The incidence of complications, according to MRI data, did not differ significantly between the groups. Notably, in the TBI group, 50% of animals (n = 4) displayed cysts; in the TBI + iAr group, cysts were detected in only one animal, which amounted to 10%. An abscess was detected in one control animal, which was the reason for excluding this animal from further analysis.

### 2.2. Changes in Body Weight

There was no significant difference in the control point of animal weight; however, it is worth noting that in the early postoperative period (1–3 days after TBI), in the SO and TBI groups, there was a loss of body weight, followed by a gradual weight increase. In contrast, in the TBI + iAr group, there was a steady tendency towards weight gain throughout all 14 days of observation (Figure 1). Body weight gain after TBI in the TBI + iAr group occurred statistically significantly faster than in the TBI group from the 7th day (Δ7 days − 0 days) of observation (*p* = 0.005), and this trend persisted on the 14th day (Δ14 days − 0 days) (*p* = 0.005).

### 2.3. Results of Neurological Examination via Limb-Placing Test (LPT)

According to the results of the neurological examination, all animals modeled for TBI showed a significant neurological deficit on the 3rd day of observation. The median score in the control group (TBI) was 2 (1; 5), and in the treated group (TBI + iAr), it was 3 (1; 9.5) (Figure 2). There was also a slight neurological deficit in the SO group, with a score of 9 (5.5; 10.5).

On day 7 of observation, there was a steady positive trend in the SO and TBI + iAr groups, with an improvement in score (ΔLPT 7d − 3d) to 3.5 (−0.75; 6) and 1 (−0.25; 3), respectively. However, in the TBI group, the neurological deficit remained at the same level—ΔLPT 7d − 3d = 0 (0; 3).

By day 14, there was a significant difference in the number of points between the groups (*p* < 0.001). A pairwise comparison showed that the sum of points was significantly higher in the TBI + Ar group than in the TBI group (*p* = 0.004), but not in the SO group (*p* = 0.210). The median was 8.5 (8; 10.3) in the TBI + iAr group, 5 (3; 6) in the TBI group, and 11 (8.75; 13.3) in the SO group (Table 1).

The increase in values on day 14 (ΔLPT 14d − 7d) was also significantly different (*p* = 0.023) in the TBI + iAr group compared to the SO and TBI, amounting to 3.5 (1; 4) vs. 0 (−1; 0.5) and 1 (−3; 4), respectively.

### 2.4. Results of Magnetic Resonance Imaging

According to the MRI of the brain, which was performed on all animals on the 14th day of observation in the TBI + iAr group, the lesion volume was significantly smaller than that in the TBI group: 40.8 (25.6; 46.1) mm^3^ vs. 60.2 (45.6; 67.6) mm^3^ (*p* = 0.0088). No significant differences were found in the TBI + iAr group when compared to the SO group (*p* = 0.4023). The TBI group differed significantly from the SO group, where the volume of the lesion caused by a craniotomy was 39.7 (29.2; 42.4) mm^3^ (*p* = 0.0019) (Figure 3).

### 2.5. Molecular Mechanisms of Neuroprotection

#### 2.5.1. Exposure to Argon Causes an Anti-Inflammatory Effect on TBI

Analysis of the signaling pathways of postconditioning at traumatic brain injury one day after the last inhalation (4th day after TBI) revealed that exposure to argon significantly suppressed the expression of the proinflammatory cytokine TNFα (*p* = 0.029) but did not have a significant effect on IL-1a (*p* = 0.343) (Figure 4A,B). The level of the transmembrane glycoprotein type 1 CD68, which is associated with macrophage activity, was also significantly lower in the TBI + iAr group than in the TBI group (*p* = 0.047) (Figure 4C).

#### 2.5.2. Exposure to Argon Affects the Antioxidant System

When studying the key signaling pathway for activation of the cell’s antioxidant defense, the redox-sensitive transcription factor (Nrf2), a significant increase in its expression was revealed after a single inhalation in the iAr 1d group compared to the Int. 1d group (*p* = 0.010). No significant difference was detected between the groups on the day after triple inhalation (*p* = 0.999) (Figure 5A).

To study the possible pro-oxidant effect of the argon-oxygen gas mixture, we analyzed the expression of one of the components of the antioxidant system of the cell membrane—NAD(P)H dehydrogenase [quinone 1] (NQO1). Exposure to argon had no effect on NQO1 (*p* = 0.886), indicating that activation of the antioxidant system is not associated with the pro-oxidant activity of the argon-oxygen gas mixture (Figure 5B).

To determine antioxidant activity in the early recovery period after TBI, the expression of heme oxygenase-1 (HO-1) was determined (Figure 5D). It was found that exposure to the argon-oxygen gas mixture decreased on average by two times (*p* = 0.029), which may indicate not so much a suppression of antioxidant activity but a decrease in the pro-oxidant processes in general.

A potential mechanism for activating the anti-ischemic effect of argon may be hypoxic preconditioning. To exclude this mechanism of activation of protective mechanisms, the effect of inhalation of an argon-oxygen gas mixture on the expression of the transcription factor inducible by hypoxia (from hypoxia-inducible factor—Hif1α) was determined. Exposure to argon had no effect on the expression of Hif1α (*p* = 0.730) (Figure 5D), which may indicate alternative ways to activate defense mechanisms.

Under conditions of traumatic brain injury, Hif1α expression in the group treated with argon decreased by an average of 30% (*p* = 0.100), which may also indicate a moderate anti-ischemic effect of argon (Figure 5C).

#### 2.5.3. Exposure to Argon Causes an Anti-Ischemic Effect

Analysis of key signaling pathways associated with preconditioning revealed that when animals were exposed to an argon atmosphere for 2 h, there was no increase in the level of the phosphorylated form of protein kinase B (pAKT) in brain tissue. However, after three days of 2 h argon inhalation, the pAKT level significantly increased in the iAr 3d group compared to the Int control 3d (*p* = 0.029), but not compared to iAr 1d (*p* = 0.229). At the same time, the total amount of protein kinase B (AKT) did not change during either the single or triple inhalation of an argon-oxygen gas mixture. However, the ratio of pAKT to AKT levels differed between the single and triple inhalations. After a single inhalation, the pAKT/AKT in the iAr group was significantly lower (*p* = 0.016) compared to that of the control group, and after three inhalations, it was significantly higher (*p* = 0.029).

When studying glycogen synthase kinase-3β (GSK3β), which is downstream of the signaling cascade, after three exposures to argon, no significant changes were detected either in GSK3β itself (*p* = 0.857) or in its phosphorylated (inactivated) form pGSK3β (*p* = 0.400). The ratio of pGSK3β to total GSK3β also was not significantly different (*p* = 0.400) (Figure 6 and Appendix A).

## 3. Discussion

The main finding that stands out in our work is the neuroprotective effect of inhaling an argon-oxygen gas mixture (Ar 70%/O_2_ 30%) in the case of open non-penetrating head injuries. Repeated (three times) two-hour inhalations of an argon-oxygen gas mixture in animals that had suffered a TBI significantly reduced the neurological deficit starting from the 7th day of observation and the volume of the lesion on the 14th day of observation.

Previously, our colleagues obtained conflicting data on therapy with an argon-oxygen breathing mixture in similar models of open traumatic brain injury. The study by Moro F. [16] showed a positive effect, while the study by Creed J. [15] showed an absence of effect in closed-head traumatic brain injury. We tested this hypothesis in a model of open-head TBI in our previous work [17], in which there was also no effect of argon treatment. A more detailed study of the factors capable of neutralizing the effects of argon revealed that one of the leading mechanisms that neutralizes the organoprotective effects of argon is the use of inhalation anesthesia with sevoflurane and isoflurane [19,30,31], which are inherently associated with neuroprotection. In the current protocol, the use of inhalational anesthetics was limited to MRI on day 14, which may have had no effect. Additionally, this study noted a more rapid recovery of the general physical condition of the animals, particularly a faster gain in body weight, which is consistent with the results of studies by other authors [21,32].

When studying the signaling pathways potentially responsible for the neuroprotective effect, special attention was paid to the molecular mechanisms underlying preconditioning. The PI3K/AKT/mTOR signaling pathway is a key regulator of neuronal cell growth, as well as axon and dendrite outgrowth, during brain development [33,34]. Activation of PI3K and the subsequent activation of AKT is stimulated by various growth factors and hormones, which regulate the activity of the mammalian target of the rapamycin (mTOR) complex, specifically molecules such as mTORC1 and mTORC2 [35]. The function of mTOR involves integrating inputs from several upstream signals related to apoptosis, the inhibition of cell proliferation, and autophagy. Neuronal mTOR controls protein synthesis in cell bodies and axons, which is crucial for cellular growth [36]. It has been demonstrated that rapamycin inhibits mTOR, reduces microglial/macrophage activation, and enhances neuronal survival [37]. Data on the effects of argon on this molecular mechanism are limited to a few studies in the field of cardioprotection [38,39]. Regarding neuroprotective effects, Zhao H. et al. (2016) illustrated a significant cytoprotective effect of argon by enhancing HO-1 expression through the PI3K-AKT signaling pathway [40], while inhibition of the phosphoinositide-3-kinase (PI3K)/AKT pathway via LY294002 attenuated the brain protection provided by argon together with hypothermia. In contrast, the study by Scheid S. et al. (2023) showed that reduced neuronal apoptosis induced by rotenone was associated with decreased AKT levels [41], alongside reduced NF-kB expression but increased ERK1/2 phosphorylation. Despite contradictions, in their review, Gardner A.J. and Menon D.K. (2018) associated the neuroprotective effect of argon with the activation of both the extracellular signal-regulated kinase (ERK) 1/2 pathway and the phosphatidylinositol 3-kinase (PI-3K)-AKT pathway [42]. Our study also showed that argon exposure increases pAKT expression only after repeated inhalation, suggesting that this effect is cumulative. Exposure to high concentrations of argon has been shown to cause a neuroprotective effect by suppressing the expression of AKT [41]; in our study, there was a slight but not statistically significant decrease in the level of this enzyme.

In the signaling cascade of mechanisms that protect against ischemic damage, one of the key factors that regulates mitochondrial pore permeability and thus, the mechanism of cell death by apoptosis, is glycogen synthase kinase 3β (GSK-3β). Glycogen synthase kinase 3β is known to regulate glycogen metabolism, protein synthesis, microtubule dynamics, cell differentiation, cell death, and apoptosis. GSK-3β signaling is responsible for neuronal death caused by numerous toxic stimuli such as beta-amyloid or apoptotic proteins such as p53 [43]. The two major signaling pathways that regulate GSK-3β activity are Wnt and AKT [44]. Abnormal activation of GSK-3β is associated with neurodegeneration and chronic neuroinfections [45,46].

In the context of craniocerebral trauma, GSK-3β has been implicated in both apoptotic cell death and neuroinflammation, and its inactivation through phosphorylation may be a potential mechanism in containing secondary brain damage. However, the effects of argon on this enzyme remain inadequately studied, with conflicting data on its combined effects with hypothermia in an in vitro model [40]. Elevated levels of GSK-3β have been linked to diseases like type II diabetes and neurodegenerative diseases [47]. Studies by Zhao H. and Tao J. have shown a correlation between GSK-3β expression, ischemic injury, and apoptosis, with argon playing a role in reducing neuronal damage [40,48]. Activation of the Wnt/β-catenin pathway can also inhibit GSK-3β expression, supporting the blood-brain barrier and mitigating damage from ischemia-reperfusion [49,50]. In our study, only a statistically insignificant increase in pGSK-3β levels, with unchanged GSK-3β levels, was observed upon exposure to argon, indicating the need for a more in-depth investigation of this pathway.

The nuclear factor associated with erythroid 2 (Nrf2) is a transcription factor that protects cells from various harmful influences. Under normal conditions, Nrf2 is located in the cytoplasm, bound to Keap1 [51,52]. When oxidative stress occurs, Nrf2 dissociates from Keap1, translocates to the nucleus, and induces the transcription of protective genes like heme oxygenase-1 (HO1), NAD(P)H dehydrogenase [quinone] 1 (NQO1), superoxide dismutase 1 (SOD1), and glutathione S-transferase (GST) through the ARE [53,54]. The ERK1/2 and PI3K-AKT pathways have also been shown to regulate the Nrf2 gene [55,56].

Studies by Goebel U. and Zhao H. et al. have reported differing effects of argon on Nrf2 activity and gene expression [57]. While Goebel U. found suppression of Nrf2 DNA-binding activity by argon, mediated through TLR2/4, Zhao H. discovered that argon increased the expression of NQO1, SOD1, and glutathione (GSH) and decreased the expression of caspase-3 through the mTOR-Nrf2 pathway mediated by ERK1/2 or PI3K-AKT [5]. However, Scheid S. et al. did not observe significant changes in Nrf2 phosphorylation following treatment with rotenone or argon [41]. Low levels of ROS may have neuroprotective effects early after ischemic injury, with their accumulation leading to cytotoxicity and Nrf2 activation [58]. The results of our study confirm this hypothesis and demonstrate that the effect of argon on a significant increase in Nrf2 expression after a single inhalation could indicate a rapid activation of this cascade, but the lack of a further increase after a triple inhalation refutes a possible cumulative effect.

In the pathogenesis of traumatic brain injury, lipid peroxidation can lead both to the death of neural tissue and to the disruption of the blood-brain barrier (BBB), which may allow activated macrophages from the systemic circulation to penetrate the brain, thereby exacerbating secondary brain damage [59]. In the study by Zhao H. et al. [5], the mechanism of antioxidant brain protection was thoroughly examined in a rat model of ischemic-hypoxic brain injury, with a focus on the role of argon as an activator of this neuroprotective mechanism. Activation of antioxidant defense mediated by Nrf2 through the phosphatidylinositol 3-kinase (PI3K)/AKT pathway (PI3K/AKT pathway) leads to increased expression of GST and NAD(P)H: quinone oxidoreductase 1 (NQO1) [60], as well as a reduction in malondialdehyde (MDA), a product of lipid peroxidation [5]. However, in our study, although a significant increase in Nrf2 expression was observed under the influence of argon, no significant increase in NQO1 expression was detected. Considering that the expression of this protein was studied in the context of an intact brain (without traumatic brain injury), this may indicate the absence of lipid peroxidation in this scenario, which supports the safety of the investigated mixture from the perspective of hyperoxia development.

Heme oxygenase (HO) is an enzyme that catalyzes the breakdown of heme. With its participation, the porphyrin ring of heme breaks down to form biliverdin, carbon monoxide (CO), and an iron atom, which is highly toxic due to its ability to react with Fenton to form hydrogen peroxide. Under conditions of traumatic brain injury, an increase in the expression of HO-1 is observed as early as one day after the injury, peaking after 3 days and returning to control levels after 7–15 days. The expression of HO-1 predominates on day 3 in the perinecrotic region and on day 7 in vascular disease. At the same time, heme deposits are gradually degraded from day 1 to day 7 and replaced by iron deposits by day 7. It is assumed that HO-1 is expressed by glial cells, mainly astrocytes (GFAP) [21].

To date, there is very little data on the effect of noble gases on the expression of HO-1. There is only one study in which an increase in HO-1 content in renal tissue was found in the first few hours after exposure to xenon, which also attenuated the damage previously associated with prolonged hypothermic storage of the graft [61]. In another large study on in vitro and in vivo models, it was shown that treatment with argon in combination with hypothermia has a pronounced neuroprotective effect under conditions of hypoxic damage to neuronal cells, which is accompanied, among other things, by an increase in the expression of HO-1, and that its inhibition levels this effect. The activation of HO-1 expression in this study was associated with the (PI3K)/AKT signaling pathway [40]. In contrast, our study found that the expression of HO-1 decreased twofold during treatment with argon. Considering the contradictory role of this enzyme in the process of heme degradation, with the formation of both the antioxidant biliverdin and its overexpression of toxic iron forms [62], as well as its multidirectional changes under the action of argon in other studies [20], our results could indicate both less active activation processes of the pro-oxidant system and depletion of this enzyme due to its increased activation in the earlier period after TBI.

The data on the role of Hif-1α in the pathogenesis of brain damage associated with the development of ischemic damage such as ischemic stroke, hemorrhage, and circulatory arrest are contradictory. It was shown in the study by Khan M. I. et al. that stabilization of Hif1α by S-nitrosylation accelerates the processes of repair and functional recovery in a model of traumatic brain injury in mice [63]. In another recent study, Xu X. et al. (2023) found, in a similar model, that increased expression of Hif1α in the acute period of TBI contributes to the loss of neurons, as well as to the activation of apoptosis and the glia, and its inhibition attenuates these processes [64].

The influence of noble gases on the expression of the protein in question also has an ambiguous nature. For example, in the study by Rizvi M. et al. (2010), preconditioning with xenon increased the expression of Hif1α, as well as pAKT and Bcl-2, while argon, neon, and krypton did not [65]. The only study that examined the neuroprotective properties of argon and the role of Hif1α in their implementation was conducted on a model of subarachnoid hemorrhage (SAH) by Höllig A. et al. (2016) [20]. In this study, researchers noted a biphasic change in the expression of Hif1α in the group receiving argon treatment, specifically a decreased expression at 6 h post-SAH, a significant increase at 24 h compared to that of the control group, and a subsequent decrease at 72 h post-SAH. However, animals receiving argon treatment displayed better survival rates and overall condition, as well as less severe neuronal deficits at 3 days, based on immunohistochemistry results. In our study, the effect of argon on Hif1α was studied both in intact brain conditions, to test the prophylactic effect of inhaling a breathing mixture, and in traumatic brain injury (TBI) conditions, to examine the assessment of the anti-hypoxic effect of argon. It was shown that argon did not have an effect on the expression of Hif1α in the intact brain, which refutes the hypothesis of its pro-hypoxic action. Further investigation into the mechanisms of argon’s neuroprotective effect in a TBI model showed that on the 4th day after TBI, the level of Hif1α in the argon group was two times lower than in the TBI group. Although the decrease in Hif1α levels under the influence of argon does not exhibit statistically significant differences, in combination with the positive clinical effect, this fact may indicate both an increase in cell resistance to hypoxia and a reduction in the amount of ischemic tissue at this specific timepoint.

Secondary damage following traumatic brain injury is closely linked to the persistent inflammatory response, which plays a key role in determining patient outcomes. Nuclear factor kappa-B (NF-κB) serves as a critical regulator of immune and inflammatory processes by controlling the expression of genes that promote inflammation and cell death, including various cytokines, chemokines, and adhesion molecules [66,67]. Studies by Ulbrich F. et al. (2014) demonstrated that treatment with argon effectively reduces the activation of NF-κB in a model of retinal ischemia-reperfusion injury [68]. This finding was later confirmed by Zhao H. et al. (2016) in a neonatal ischemia-hypoxia model [40]. Additionally, the protective effect of argon against cell death was diminished in the presence of an NF-κB inhibitor called Bay11-7082 [69].

It is known that Toll-like receptors 2 and 4 (TLR2/4) play a key role in activating the NF-κB signaling pathway [70,71]. Argon treatment has been found to effectively suppress the upregulation of TLR2/4, NF-κB, and various pro-inflammatory factors like IL-1α, IL-1β, IL-6, TNFα, and iNOS after trauma However, this inhibitory effect was reversed by TLR2/4 inhibitors [57,69]. Interestingly, when TLR2/4 inhibition was combined with OxPAPC pretreatment, the suppression of NF-κB by argon was not significantly affected, suggesting that argon may be targeting alternative pathways to regulate NF-κB, such as the TNFR/IL-1R-TAK1-NF-κB signaling pathway [41].

One of the main potential effects expected from argon treatment in traumatic brain injury (TBI) is the suppression of aseptic inflammation, which is at the core of further neuronal death in the early stages of injury, worsening the long-term consequences of the trauma. In several studies, argon has shown anti-inflammatory effects by suppressing factors that support inflammation. For example, in several studies, the suppression of the infiltration of the injury zone by macrophages circulating in the bloodstream and the activation of microglia has been noted [16,28]. In our study, argon reduced the expression of CD-68 positive cells in the injury zone by two times, bringing their levels closer to those in the intact brain. The pro-inflammatory cytokine TNFa also significantly decreased, indicating the regulation by argon of the inflammatory response in the early post-traumatic period [72] (Figure 7).

We would like to point out that this study has limitations that need to be considered when interpreting the results. Firstly, the treatment regimen (frequency and duration of inhalations) is suboptimal and needs further investigation to determine the minimum and maximum therapeutic effects for this condition. Additionally, the study of molecular mechanisms is partial, providing a direction for further research but not a comprehensive understanding of the phenomena observed. Nevertheless, we believe that our findings may be valuable for other researchers in this field.

## 4. Materials and Methods

### 4.1. Experimental Animals

Experiments were performed on male Wistar rats weighing 330–380 g (n = 61), maintained on a 12/12 h light/dark cycle and at a temperature of 22 ± 2 °C, with free access to food and water. In the experiment, an argon-oxygen gas mixture (Ar 70%/O_2_ 30%, “ArgOx 70”, Akela-N LLC, Moscow, Russia) and a nitrogen-oxygen gas mixture (N_2_ 70%/O_2_ 30%, “Nitrogen-Oxygen 70/30”, Akela-N LLC, Moscow, Russia) were used. The study protocol was approved by the Local Ethics Committee of the Federal Research and Clinical Center of Intensive Care Medicine and Rehabilitology of the Russian Federation (protocol 2/24/1, dated 18 June 2024) (Figure 8). The experiments were conducted in accordance with the requirements of Directive 2010/63/EU of the European Parliament and the Council of the European Union for the Protection of Animals Used for Scientific Purposes.

The animals were randomly divided into five groups according to the extent of the surgery performed:-Sham-operated animals that received anesthesia and preparatory procedures, without TBI and inhalations (SO group), n = 6;-Control group treated with TBI + N_2_ 70%/O_2_ 30% inhalation three times (TBI group), n = 16;-Experimental group treated with TBI + Ar 70%/O_2_ 30% inhalation three times (TBI + iAr group), n = 17;-Intact animals that received inhalations with a nitrogen-oxygen gas mixture N_2_ 70%/O_2_ 30% only once (group Int. 1d, n = 6) or three times (group Int. 3d, n = 5);-Intact animals that received inhalations with an argon-oxygen gas mixture Ar 70%/O_2_ 30% only once (group iAr 1d, n = 6) or three times (group iAr 3d, n = 5).

### 4.2. Modeling of TBI

After intraperitoneal anesthesia with chloral hydrate and topical application of a long-acting local anesthetic, bupivacaine [19,73], TBI was induced via the dosed contusion injury method [74,75]. The skin on the rat’s head was shaved in the surgical field and treated with the antiseptic chlorhexidine 0.05%. The rat was placed in a stereotaxic frame. A skin incision was made along the sagittal suture, and a hole was drilled into the parietal and frontal cranial bone above the left hemisphere using a 5 mm diameter drill to locate the sensorimotor cortex at coordinates 2.5 mm lateral to the sagittal suture and 1.5 mm caudal to the bregma [76]. The injury device was positioned so that the firing pin was above the dura mater. To cause an injury, a 50 g weight was dropped on the striker with a diameter of 4 mm from a height of 10 cm to a depth of 1 mm [77]. The skin was sutured with Vicryl No. 4, and the surgical area was treated with a 5% solution of brilliant green. Before the animal recovered from anesthesia, the body temperature was maintained at 37 ± 0.5 °C [78,79] using an infrared lamp and a rectal body temperature sensor connected to a thermostat [75]. A hole was also drilled in the skull of the sham-operated animals, without causing any damage.

Mortality in the animal groups was assessed 24 h after TBI and on the 7th and 14th days after TBI.

### 4.3. Exposure to Argon

The animals were placed in a transparent plastic chamber with a volume of 35 L into which a fresh gas mixture (N_2_ 70%/O_2_ 30%—TBI group, Int. 1d and 3d; Ar 70%/O_2_ 30%—TBI + iAr, iAr 1d and 3d) was constantly fed at a flow rate of 0.5 L per animal. No more than five animals in the same group were in the chamber at the same time. In the TBI and the SO groups, inhalations were administered immediately after awakening (on average 90 ± 15 min after TBI). The animals also received anesthesia with 50 mg/kg s/c paracetamol after the operation [80,81].

In the TBI and TBI + iAr groups, inhalations were conducted for 2 h for 3 days (on the day of injury (day 0), 24 h after TBI (day 1), and 48 h after TBI (day 2)), but the SO group received no inhalations. After the end of the exposure period, the general condition of the animals was assessed (level of wakefulness, mobility). The animals were then moved into cages, with free access to water and food. In the early postoperative period (the first three days after TBI), prolonged analgesia was administered with a liquid solution of paracetamol in a drinking bowl at a maximum dose of 337 mg/kg/day [80].

In the groups used for investigating the molecular mechanisms of the neuroprotective effects of inhalation, 2 h inhalations were performed once in the Int. 1d and iAr 1d groups or three times in the Int. 3d and iAr 3d groups, in accordance with the previously described methods.

### 4.4. Assessment of Neurological Status

Body weight measurements and assessments of the neurological status of the animals were conducted on day prior to the experiment (M0), and 24 h (M1), 3 days (M3), 7 days (M7), and 14 days (M14) after the start of the experiment. Neurological status was assessed on the 3rd day of observation (LPT-3), the 7th day of observation (LPT-7), and the 14th day of observation (LPT-14) after TBI.

The limb-placing test (LPT) was conducted using a protocol based on the technique outlined by De Ryck et al. [82] and modified by J. Jolkkonen et al. [83]. Prior to the test, the rats were handled for one week. The test consisted of seven trials to assess the sensorimotor integration of both the forelimbs and hindlimbs in response to tactile, proprioceptive, and visual stimuli. Each trial was assigned a score as follows: 2 points for normal task performance, 1 point for delayed (>2 s) or incomplete task performance, and 0 points for failure to perform the task.

### 4.5. Brain MRI

On the 14th day post-TBI, an MRI study was conducted using a tomograph with a 7 Tesla magnetic field and a gradient system of 105 mT/m (BioSpec 70/30, Bruker, Germany). The animals were anesthetized with isoflurane (1.5–2%) and positioned using a system of stereotaxis and thermoregulation [84].

A standard protocol for studying the rat brain was followed, which included obtaining T2-weighted images. A linear transmitter with an internal diameter of 72 mm was used to transmit a radio frequency (RF) signal, and a surface receiving coil for the rat brain was used to detect the RF signal. The pulse sequences (PS) used were RARE—spin echo-based PS—with the following parameters: TR = 6000 ms, TE = 63.9 ms, slice thickness 0.8 mm with a step of 0.8 mm, matrix size 256 × 384, and resolution of 0.164 × 0.164 mm/pixel. The scanning time for one animal was approximately 25 min. The degree of brain damage was assessed using graphical analysis of MRI images, with calculation of the volume of the damaged area of the brain. To do this, the damage area of each slice, in mm^2^, was calculated using a series of MRI images using the ImageJ program (National Institutes of Health image software, Version 1.8.0, Bethesda, MD, USA). The damage volume was calculated according to a previously published methodology [85].

### 4.6. Western Blot Analysis

The day after the last inhalation, the rats were decapitated under anesthesia, after which the cerebral hemispheres were craniotomized and removed. The brain hemispheres were homogenized in PBS at pH 7.4 (Amresco, Cleveland, OH, USA) supplemented with 1 mM protease inhibitor PMSF (Amresco, USA). The total protein concentration was determined using a commercial bicinchoninic acid kit (Sigma, St. Louis, MO, USA). Brain homogenate samples were placed on gradient (5–20%) Tris-glycine polyacrylamide gels (10 μg of total protein per lane). After electrophoresis, the gels were transferred to PVDF membranes (Amersham Pharmacia Biotech, Newcastle, UK). The membranes were blocked with 5% (*w*/*v*) skim milk (SERVA, Heidelberg, Germany) in PBS supplemented with 0.05% (*v*/*v*) Tween 20 (Panreac, Barcelona, Spain) and then incubated with the following primary antibodies: rabbit polyclonal antibodies against Akt 1:1000 (#9272, Cell Signaling, Danvers, MA, USA), against phosphorylated Akt 1:700 (#4060, Cell Signaling, USA), and against phosphorylated GSK3b 1:500(#9336, Cell Signaling, USA); mouse polyclonal antibodies against GSK3α/β 1:1000 (#sc-9271, Santa Cruz Biotech, Dallas, TA, USA); rabbit polyclonal antibodies against Nrf2 1:1000 (#ab137550, Abcam, Cambridge, UK), anti-NFkB (p50) 1:1000 (#sc-114, Santa Cruz Biotech, USA), anti-Hif1α 1:1000 (#AF1009, Affinity, Zhenjiang Shi, China), anti-CD68 1:1000 (#DF7518, Affinity, China), anti NQO1 1:1000 (ab34173, Abcam, UK), and anti-heme oxygenase-1 (HO-1) 1:1000 (ab13243, Abcam, UK); mouse monoclonal antibodies against IL-1a 1:1000 (#GTX10747, GeneTex, Alton Pkwy Irvine, CA, USA), anti-TNFα 1:250 (#88-7340-88, eBioscience, San Diego, CA, USA), and anti-β-actin 1:2000 (#A2228, Sigma, USA).

The membranes were stained with secondary antibodies against rabbit IgG or mouse IgG conjugated with horseradish peroxide at a dilution of 1:5000 (both from IMTEK, Moscow, Russia). Specific bands were visualized using the Advansta Western Bright^TM^ ECL kit (Advansta, San Jose, CA, USA). Detection was performed on a V3 Western Blot Imager system (BioRad, Hercules, CA, USA), and band density was measured using Image Lab software (BioRad, Version 5.1, Hercules, CA, USA). β-actin was used as an internal control, and the total proteins AKT and GSK3α/β were used to normalize the signal of phosphorylated forms of proteins.

### 4.7. Statistical Analysis

Statistical data processing was performed using SPSS Statistics (IBM SPSS Statistics for Windows, Version 27.0.1, Armonk, NY, USA: IBM Corp.) and GraphPad Prism (GraphPad Software, Version 8.0.1, Boston, MA, USA). The normality of the distribution of the trait in the samples was assessed using the Shapiro-Wilk test. The data are presented as Me (Q1; Q3), where Me is the median value, Q1 is the first quartile (25th percentile), and Q3 is the third quartile (75th percentile). The Kruskal-Wallis test with post hoc analysis (Benjamini-Krieger-Yekutieli method) and the Mann-Whitney U test (Wilcoxon rank-sum test) were applied for continuous variables. The Chi-square test or Fisher’s exact test (in cases where the outcome frequency was less than 10%) were used to compare categorical variables between independent groups. The statistical significance level was set at *p* < 0.05.

## 5. Conclusions

Inhalations of an argon-oxygen gas mixture (Ar 70%/O_2_ 30%) with a 2 h exposure duration, repeated three times, demonstrate a neuroprotective effect on open traumatic brain injuries. This is supported by functional neurological status assessments (LPT test) and instrumental MRI data, which show a significant reduction in the lesion area on the 14th day of observation. The molecular mechanisms underlying argon’s neuroprotective effect in the context of preconditioning are mediated by the activation of the AKT and Nrf2 signaling pathways. In postconditioning, argon inhalation suppresses neuroinflammation by reducing the expression of CD68 and TNFα.

## Figures and Tables

**Figure 1 ijms-25-12673-f001:**
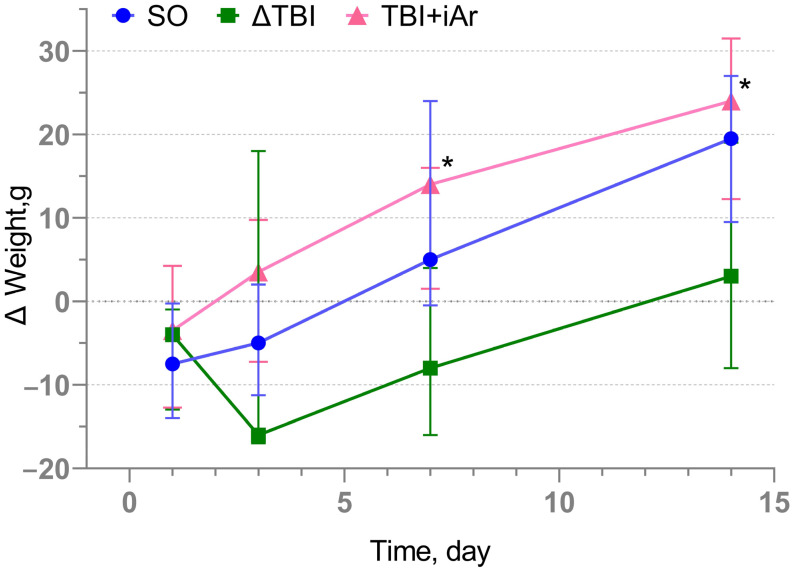
Analysis of body weight changes after TBI modeling (score increase compared to D0). The data are presented as median and inter quartile range (Q1; Q3). * *p* < 0.05 vs. TBI group according to the Kruskal-Wallis test with the two-stage linear step-up procedure of Benjamini, Krieger, and Yekutieli.

**Figure 2 ijms-25-12673-f002:**
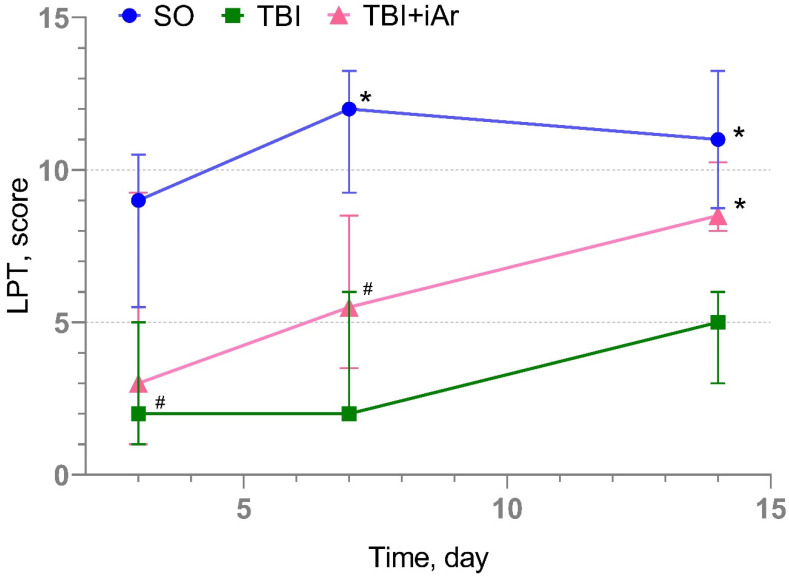
Dynamics of changes in neurological status in groups according to the results of the LPT. The data are presented as median (inter quartile range), * *p* < 0.05 vs. TBI group, # *p* < 0.05 vs. SO group according to the Kruskal-Wallis test with the two-stage linear step-up procedure of Benjamini, Krieger, and Yekutieli.

**Figure 3 ijms-25-12673-f003:**
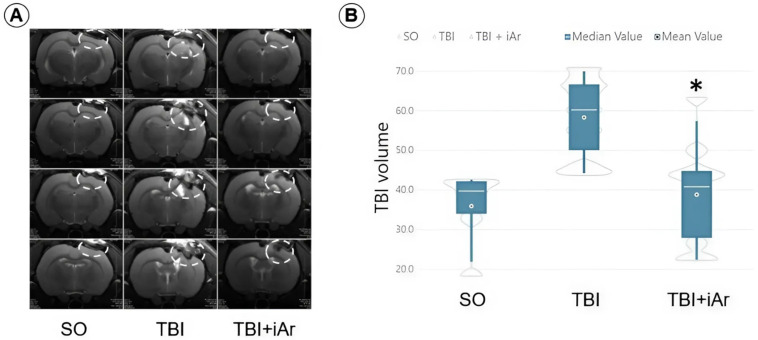
MRI of the brain: (**A**) T2-weighted coronal MRI image (A-NN12,14) of an animal group; (**B**) volume damage zones on the 14th day of observation; data are presented as median and inter quartile range (Q1; Q3). * *p* < 0.05 vs. TBI group according to the Kruskal-Wallis test with the two-stage linear step-up procedure of Benjamini, Krieger, and Yekutieli.

**Figure 4 ijms-25-12673-f004:**
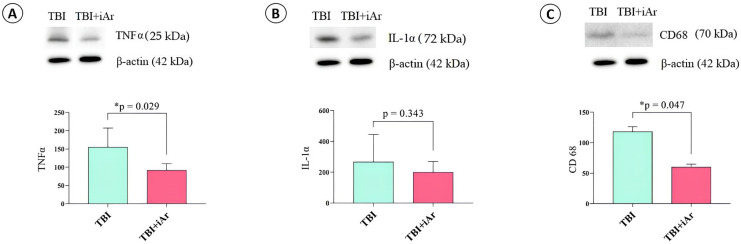
Anti-inflammatory activity of argon. Changes in signaling pathways in the rat cerebral cortex due to TBI and argon inhalation. Representative immunoblots and densitometry for (**A**) TNFα, (**B**) IL-1α, and (**C**) CD68 signals were normalized to β-actin as the loading control. The observed molecular weights of the proteins are labeled. The data are presented as median and inter quartile range (Q1; Q3). * *p* < 0.05 vs. TBI group, according to the Mann-Whitney U-test.

**Figure 5 ijms-25-12673-f005:**
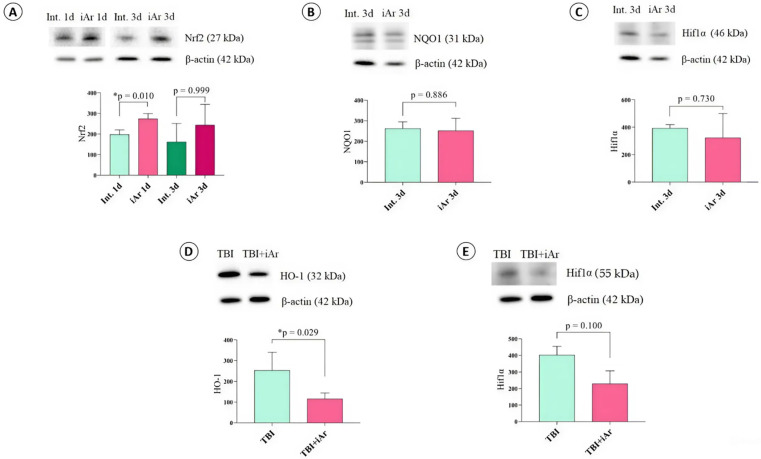
The effect of argon on the antioxidant system and factors induced by hypoxia is studied. Representative immunoblots and densitometry for (**A**) Nrf2, (**B**) NQO1, (**C**) Hif1α after argon exposure and (**D**) HO-1 and (**E**) Hif1α by TBI and argon inhalation are shown. Signals were normalized to β-actin as the loading control. The observed molecular weights of the proteins are labeled. The data are presented as median and inter quartile range (Q1; Q3). * *p* < 0.05 vs. TBI group according to the Mann-Whitney U-test.

**Figure 6 ijms-25-12673-f006:**
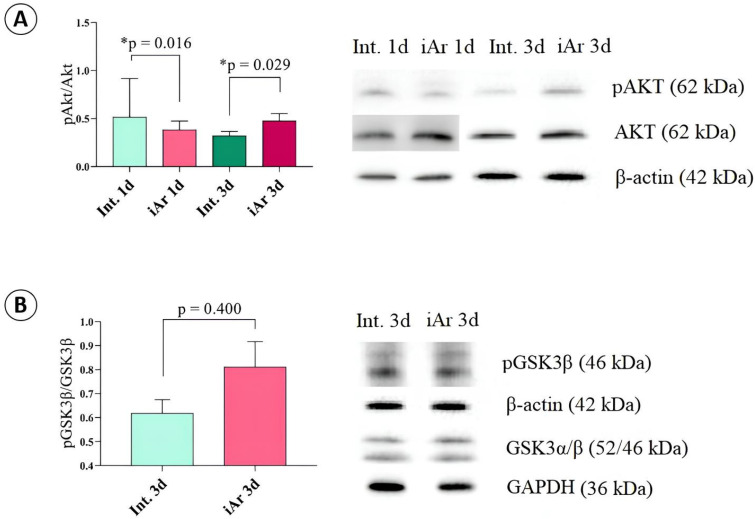
Influence of argon on preconditioning: representative immunoblots and the ratio of densitometry results of phosphorylated forms of proteins to the total (**A**) pAKT/AKT and (**B**) pGSK/GSK after argon exposure. Signals were normalized to β-actin or GAPDH as the loading control. The observed molecular weights of the proteins are labeled. The data are presented as median and inter quartile range (Q1; Q3). * *p* < 0.05 vs. the TBI group based on the Mann-Whitney U-test.

**Figure 7 ijms-25-12673-f007:**
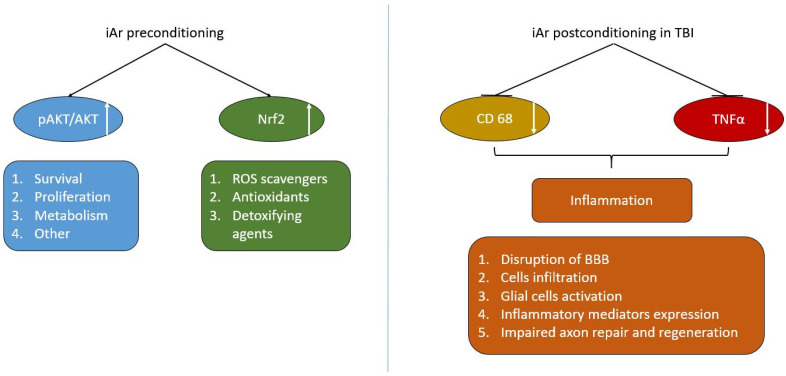
Scheme of the implementation of the neuroprotective effect of inhalation of an argon-oxygen gas mixture in TBI. This figure illustrates a general outline of the mechanism of the neuroprotective effect of argon under conditions of an intact brain (preconditioning) and traumatic brain injury (postconditioning). Inhalation of an argon-oxygen breathing mixture in the intact brain increases the expression of the phosphorylated form of Akt, which in turn activates processes of cell survival, proliferation, and metabolism in the brain cells [34]. Activation of Nrf2 expression protects brain cells from the action of reactive oxygen species (ROS) [52], increases antioxidant protection, and detoxification function. Under the conditions of traumatic brain injury, argon inhalation reduces the expression of CD68 and TNFα, indicating suppression of neuroinflammation, which helps to reduce damage to the blood-brain barrier (BBB), cell infiltration, astrocyte activation, and the decrease in the release of inflammatory mediators, as well as to improve axon repair and regeneration in the perifocal zone [23,72].

**Figure 8 ijms-25-12673-f008:**
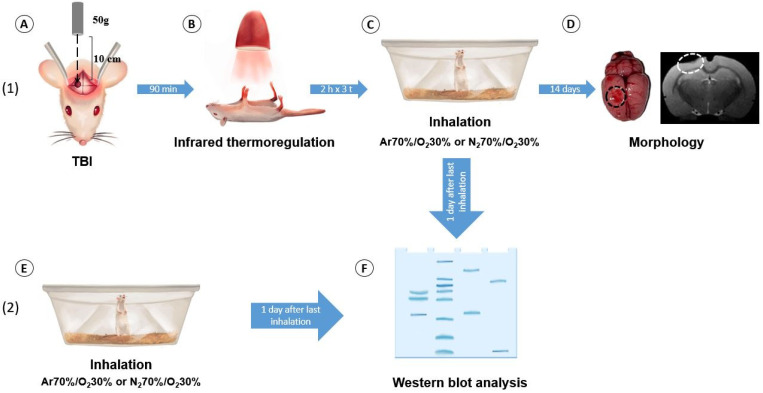
The scheme of the experimental protocol with the evaluation of the effect of argon on rats with traumatic brain injury: (**A**) induction of craniocerebral trauma in a rat using a free-falling load model; (**B**) maintenance of the rat’s body temperature using an infrared lamp; (**C**,**E**) inhalation by the rats of a gas mixture in a special chamber; (**D**) evaluation of the volume of damage to the cerebral cortex; (**F**) Western blot analysis of the signaling mechanisms of neuroprotection.

**Table 1 ijms-25-12673-t001:** Results of neurological examination of via the limb-placing test.

Parameter	Group			*p*-Value			
	SO	TBI	TBI + iAr	K-W test	SO vs. TBI	SO vs. TBI + iAr	TBI vs. TBI + iAr
LPT 3 d, points	9 (5.5; 10.5)	2 (1; 5)	3 (1; 9.25)	0.0399	0.0131 *	0.0612	0.4006
LPT 7 d, points	12 (9.25; 13.3)	2 (2; 6)	5.5 (3.5; 8.5)	0.0031 *	0.0008 *	0.0163 *	0.2068
LPT 14 d, points	11 (8.75; 13.3)	5 (3: 6)	8.5 (8; 10.3)	0.0005 *	0.0002 *	0.2090	0.0036 *
ΔLPT 7d − 3d, points	3.5 (−0.75; 6)	0 (0; 2)	1 (−0.25; 3)	0.4349	0.2137	0.3160	0.7242
ΔLPT 14d − 3d, points	2 (−1; 6)	2 (1; 4)	5.5 (0.75; 6.25)	0.3210	0.8636	0.2625	0.1714
ΔLPT 14d − 7d, points	0 (−1; 0.5)	1 (−3; 4)	3.5 (1; 4)	0.0232 *	0.1827	0.0064 *	0.1761

Note: The data are presented as median and inter quartile range (Q1; Q3). * *p* < 0.05 according to the Kruskal-Wallis test with the two-stage linear step-up procedure of Benjamini, Krieger, and Yekutieli.

## Data Availability

Data can be provided on request.

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
