# Peer review of "Positive Effects of Argon Inhalation After Traumatic Brain Injury in Rats"

_ijms, 2024, doi:10.3390/ijms252312673_

Round 1

Reviewer 1 Report

Comments and Suggestions for Authors

Major Concerns

Results

The TBI group showed a mortality rate of 18.9%, which suggests that the CCI model may cause pain in the animals. Did the study team administer any analgesics to mitigate this pain? If so, what type of analgesic was used?

Discussion

You provided a sufficient discussion on the BBB after TBI, yet the study did not evaluate BBB integrity.

The study team emphasized the activation of the PI3K/Akt/mTOR pathway triggered by Argon in the discussion, but only Akt and pAkt were measured in the study. Why were other components of the pathway not investigated?

In Lines 255256, you mention that Argon may decrease Akt expression. However, Figure 6A shows results that contradict this claim: the band signals for iAr 1d and iAr 3d are higher than the control after Argon treatment.

Methods

Please include additional details regarding your TBI model. What was the impact depth? What was the diameter of the striker?

Minor Concerns

Line 47: Grammar error in "In another day".

Line 61: There are two periods at the end of the sentence.

Line 70: Typo in "The animal mutilates itself)".

Line 350: "As in the study by Khan" is incomplete.

Many errors such as O2, N2, mm3 etc.

HIF-1α is mistakenly written as HIF1a multiple times in the discussion.

Reviewer 2 Report

Comments and Suggestions for Authors

I appreciate the authors for presenting this research article, which emphasizes the effects of Argon gas in experimental TBI rats. This study is well-designed and organized, with robust results that support the hypothesis. My comments are as follows:

 1. The results demonstrate only the association between Argon gas and various TBI-induced parameter changes, with all parameters occurring simultaneously. However, interactions among these parameters cannot be identified.

2. Figure 7 is well-presented. However, it would be helpful to add a graphical abstract illustrating the mechanisms of Argon's neuroprotective effects based on the study’s findings.

3. The limitations of the current study should be discussed.

Round 2

Reviewer 1 Report

Comments and Suggestions for Authors

All the concerns have been addressed.